# SARS-CoV-2 spike-specific memory B cells express higher levels of T-bet and FcRL5 after non-severe COVID-19 as compared to severe disease

Raphael A. Reyes[1], Kathleen Clarke[1], S. Jake Gonzales[1], Angelene M. Cantwell[1], Rolando Garza[1], Gabriel Catano[2], Robin E. Tragus[2], Thomas F. Patterson[2,3], Sebastiaan Bol[1], Evelien M. Bunnik[1]*

1 Department of Microbiology, Immunology and Molecular Genetics, Long School of Medicine, The University of Texas Health Science Center at San Antonio, San Antonio, Texas, United States of America, 2 Department of Medicine, Division of Infectious Diseases, The University of Texas Health Science Center at San Antonio, University Health System, San Antonio, Texas, United States of America, 3 The South Texas Veterans Health Care System, San Antonio, Texas, United States of America

* bunnik@uthscsa.edu

**Data Availability Statement:** All relevant data are within the paper and its Supporting information files.

## Abstract

SARS-CoV-2 infection elicits a robust B cell response, resulting in the generation of long-lived plasma cells and memory B cells. Here, we aimed to determine the effect of COVID-19 severity on the memory B cell response and characterize changes in the memory B cell compartment between recovery and five months post-symptom onset. Using high-parameter spectral flow cytometry, we analyzed the phenotype of memory B cells with reactivity against the SARS-CoV-2 spike protein or the spike receptor binding domain (RBD) in recovered individuals who had been hospitalized with non-severe (n = 8) or severe (n = 5) COVID-19. One month after symptom onset, a substantial proportion of spike-specific IgG⁺ B cells showed an activated phenotype. In individuals who experienced non-severe disease, spike-specific IgG⁺ B cells showed increased expression of markers associated with durable B cell memory, including T-bet and FcRL5, as compared to individuals who experienced severe disease. While the frequency of T-bet⁺ spike-specific IgG⁺ B cells differed between the two groups, these cells predominantly showed an activated switched memory B cell phenotype in both groups. Five months post-symptom onset, the majority of spike-specific memory B cells had a resting phenotype and the percentage of spike-specific T-bet⁺ IgG⁺ memory B cells decreased to baseline levels. Collectively, our results highlight subtle differences in the B cells response after non-severe and severe COVID-19 and suggest that the memory B cell response elicited during non-severe COVID-19 may be of higher quality than the response after severe disease.

**Funding:** This work was supported by a COVID-19 pilot award from the UT Health Long School of Medicine (10009547 to E.M.B.). R.A.R. was supported by Translational Science Training award TL1 TR002647. Data were generated in the Flow Cytometry Shared Resource Facility, which is supported by UT Health, NIH-NCI P30 CA054174-20 (CTRC at UT Health) and UL1 TR001120 (CTSA grant). The funders had no role in study design, data collection and analysis, decision to publish, or preparation of the manuscript.

**Competing interests:** The authors have declared that no competing interests exist.

## Introduction

Severe acute respiratory syndrome coronavirus 2 (SARS-CoV-2) is responsible for the global coronavirus disease 2019 (COVID-19) pandemic resulting in more than 5 million deaths reported worldwide as of December 2021 [1]. Highly efficacious vaccines have limited SARS-CoV-2 transmission and significantly reduced morbidity and mortality in regions of the world with access to these vaccines. However, the majority of the world's population lives in areas with low vaccination rates and remain at higher risk of SARS-CoV-2 infection and COVID-19. Both vaccination and natural infection elicit immunological protection against SARS-CoV-2 (re-)infection. Although mRNA vaccination elicits higher antibody titers and more diverse antibody responses against the SARS-CoV-2 spike protein than natural infection [2, 3], results from the first longitudinal studies in unvaccinated individuals with prior COVID-19 suggest that naturally acquired immune responses are maintained for at least a year after infection and that these responses protect from subsequent re-infection [4–11]. Because many people remain unvaccinated, it will be important to understand the immune response elicited by natural infection and whether the durability of naturally acquired immunity is influenced by the severity of disease.

Although SARS-CoV-2 infection elicits robust responses in both the T cell and B cell arms of the adaptive immune system [12, 13], the majority of research efforts have focused on B cell and antibody responses, which are thought to be critical for the control of infection and protection against re-infection. Humoral immune responses against pathogens consist of multiple components. The early B cell response is dominated by short-lived antibody-secreting cells, called plasmablasts. Simultaneously, other B cell populations undergo selection for high affinity antigen-binding in germinal centers of the secondary lymphoid organs, and can differentiate into long-lived, antibody-secreting plasma cells that migrate to the bone marrow or into memory B cells that remain in the circulation (reviewed in [14]). COVID-19 patients rapidly generate potent neutralizing IgG antibodies against the spike protein [15, 16]. Anti-spike antibody titers peak within the first two months post-infection, decline in the subsequent 3–4 months, and then plateau at titers higher than those detected in pre-pandemic samples, but are lower than anti-spike antibody titers elicited by vaccination [3, 5, 7, 17]. These phases of the anti-spike IgG profile in the circulation are the result of early short-lived plasmablast responses, followed by the secretion of antibodies by longer-lived bone marrow plasma cells [5]. In addition, spike-specific IgG$^+$ memory B cells are maintained or even increase in numbers for at least six months to one year following SARS-CoV-2 infection [6–8, 18]. Analysis of monoclonal antibodies obtained from these memory B cells revealed continued evolution of the B cell response over time, as evidenced by higher levels of somatic hypermutation, resulting in increased binding affinity and neutralizing capacity [7, 19, 20].

In recent years, novel subsets of activated B cells and memory B cells have been identified. These novel B cell subsets have mainly been defined by their phenotype, that is, the collection of markers expressed on their cell surface or intracellularly, and are thought to possess either protective or pathogenic functions in the immune response. For example, several subsets of double negative (DN; CD27$^-$ IgD$^-$) B cells have been defined in recent years [21, 22]. Type 1 double negative cells (DN1 cells), defined as DN cells that express CXCR5 or CD21 and lack CD11c and FcRL5, are transcriptionally similar to class-switched memory B cells and seem to be part of a functional B cell response [21, 23]. On the other hand, type 2 DN cells (DN2 cells), defined as CD11c$^+$/FcRL5$^+$ and CXCR5$^-$/CD21$^-$ DN cells, are thought to arise from the activation of naïve B cells outside of the B cell follicle in the lymph node. Type 3 DN cells (DN3 cells), defined as CD11c$^-$/FcRL5$^-$ and CXCR5$^-$/CD21$^-$ DN cells, have also been associated with this extrafollicular response and may be DN2 cell precursors [22, 23]. DN2 cells are more

abundant in patients with autoimmune disorders, such as systemic lupus erythematosus, and are thought to differentiate into auto-antibody-secreting cells. DN2 cells are also observed in the circulation of patients with severe COVID-19 [22], an observation that has been linked to lack of germinal center formation, presumably leading to an impaired B cell response [24].

Interestingly, memory B cells (CD27[+] IgD[-]) expressing some of the same markers as DN2 cells, including CD11c and FcRL5, as well as the transcription factor T-bet, have been described as part of normal, functional immune responses [25, 26]. Specifically, these cells have been implicated in long-lived B cell memory after tetanus toxoid and influenza virus vaccination [27, 28]. In recovered COVID-19 patients, memory B cells present 1–3 months after symptom onset are also found to express FcRL5 and T-bet, suggesting that these markers may delineate a subset of long-lived memory B cells against SARS-CoV-2 [13, 29]. However, it is unclear whether disease severity influences the development of memory B cells that express T-bet or FcRL5, and whether the phenotype of memory B cells changes with the continued evolution of this B cell compartment in the months following SARS-CoV-2 infection. A more detailed analysis of B cell subsets in recovered COVID-19 patients will therefore be needed to acquire insight into the early B cell response and the evolution of memory B cells over time.

Here, we performed spectral flow cytometry to characterize the phenotype of SARS-CoV-2-specific B cells in unvaccinated patients who recovered from non-severe or severe COVID-19. We focused on two comparative analyses. First, we compared the phenotype of spike-specific B cells between convalescent patients who were hospitalized with non-severe and severe disease to determine whether the B cell response developed differently in these two groups. Second, we compared the phenotype of memory B cells shortly after recovery to that at five months post-symptom onset to analyze the ongoing evolution of the B cell response. The results from this study provide insight into naturally acquired B cell memory against SARS-CoV-2 and a better understanding of the characteristics of durable B cell immunity.

## Results

### Study criteria

Study participants (n = 16) were enrolled in the Adaptive COVID-19 Treatment Trial (ACTT)-1 or ACTT-2 clinical trials, which were designed to evaluate the effect of remdesivir or baricitinib plus remdesivir for the treatment of COVID-19 in hospitalized patients [30, 31] and co-enrolled in the University of Texas Health San Antonio COVID-19 Repository, which provided samples for this study. Participants identified predominantly as white (94%) and Hispanic (75%), and all except one individual had one or more comorbidities (S1 Table). Participants were classified as non-severe (n = 11) or severe (n = 5) COVID-19 cases based on their worst ordinal score for disease severity during hospital stay. In summary, non-severe cases did not require supplemental oxygen (score = 4) or required supplemental oxygen or non-invasive ventilation (score = 5 or 6), while severe cases needed invasive mechanical ventilation (MV) or extracorporeal membrane oxygenation (ECMO) (score = 7). The definition of severe disease was made based on the need for MV or ECMO, because this distinguishes the most critical patients, who are the most likely to develop impaired immune responses [24, 32]. Blood was collected after discharge from the hospital (a median of 30 days after symptom onset for non-severe cases (n = 8) and 32 days after symptom onset for severe cases (n = 5)) (Fig 1). For participants with non-severe disease, additional samples were collected at an earlier time point (a median of 18 days post-symptom onset; n = 9) and at a follow-up visit approximately 5 months post-symptom onset (median, 147 days; n = 7) based on availability and consent. Peripheral

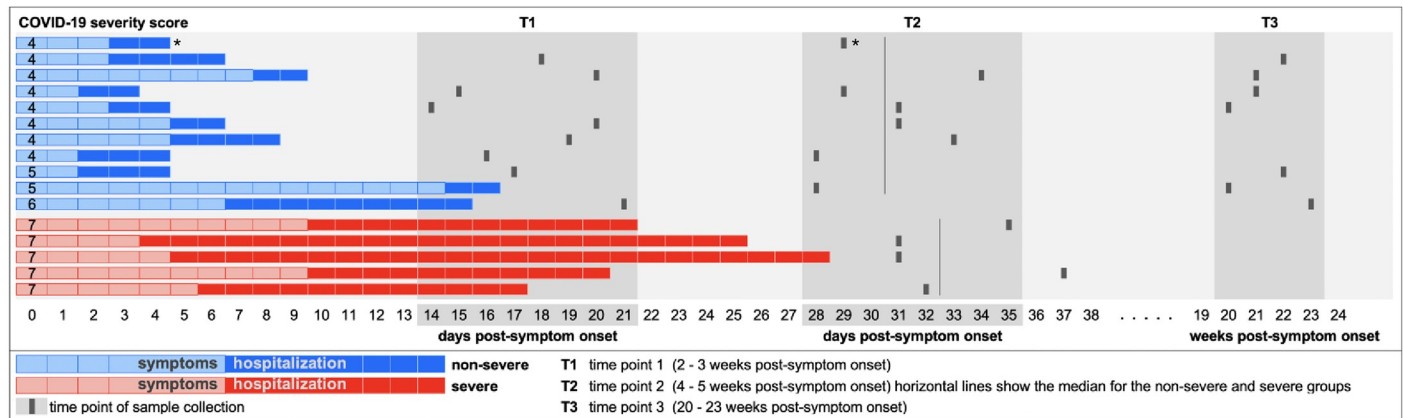

**Fig 1. Timing of sample collection.** Hospitalized COVID-19 patients were recruited for the study and sampled after hospital discharge. Samples from T2 were used for the comparison of B cell responses between individuals who recovered from non-severe and severe COVID-19 (Figs 2–4). All samples at T2 were collected 4–5 weeks post-symptom onset (PSO), with the exception of one severe donor who was sampled just after 5 weeks PSO (37 days). Longitudinal analysis of samples collected at T1–T3 could only be performed for individuals who experienced non-severe disease, because the majority of severe patients were still hospitalized at T1 and we were unsuccessful in recruiting recovered severe patients for a follow-up blood draw at T3. The donor marked with * was included in the analysis of B cell subsets (Fig 2) and the percentage of spike-specific B cells (Fig 3C), but could not be included in all subsequent analyses because only a small number of spike-specific B cells (n = 6) was detected in this sample.

blood mononuclear cells (PBMCs) from healthy donors (n = 3) collected before the start of the pandemic (2018 or early 2019) were used as negative controls.

## The proportions of B cell subsets in convalescent patients are similar between non-severe and severe COVID-19 cases

Severe COVID-19 is associated with an extrafollicular B cell response characterized by a high percentage of DN2 B cells in the circulation, similar to what is observed in patients with auto-immune disorders [21, 22]. Previous data suggest that the percentage of DN2 B cells in patients with severe COVID-19 had returned to normal levels approximately 1.5 months after hospital discharge [33]. To determine whether COVID-19 severity affects the distribution of B cell subsets shortly after recovery (four to five weeks after symptom onset), we analyzed the relative proportions of B cell subsets individuals who recovered from non-severe (n = 8) or severe (n = 5) disease by spectral flow cytometry using an antibody panel against 19 cell surface and intracellular markers (S2 Table). B cells were first gated on live, CD19$^+$ CD20$^+$ CD38$^{-/lo}$ cells to exclude transitional B cells and antibody-secreting cells, and were subsequently divided into naïve (CD27$^-$ IgD$^+$), unswitched memory (CD27$^+$ IgD$^+$), resting switched memory (CD27$^+$ IgD$^-$ CD21$^+$), activated switched memory (CD27$^+$ IgD$^-$ CD21$^-$) and double negative (DN; CD27$^-$ IgD$^-$) B cells (Fig 2A). Apart from a significant increase in the percentage of unswitched memory B cells in individuals who had experienced severe disease compared to those with non-severe COVID-19, no differences in the distribution of these B cell subsets were observed between the two groups of recovered COVID-19 patients (Fig 2B and 2C, S1A Fig). Among DN cells, DN1 (FcRL5$^-$ CXCR5$^+$) B cells were the dominant subset in both groups of recovered COVID-19 patients, and we did not detect significant differences in the frequencies of DN1, DN2 (FcRL5$^+$ CXCR5$^-$), and DN3 (FcRL5$^-$ CXCR5$^-$) B cells between the two groups (Fig 2D, S1B Fig). In combination with previous reports [22, 33], these results suggest that expansion of the DN2 B cell population during severe COVID-19 might be transient and that these cells disappear soon after recovery.

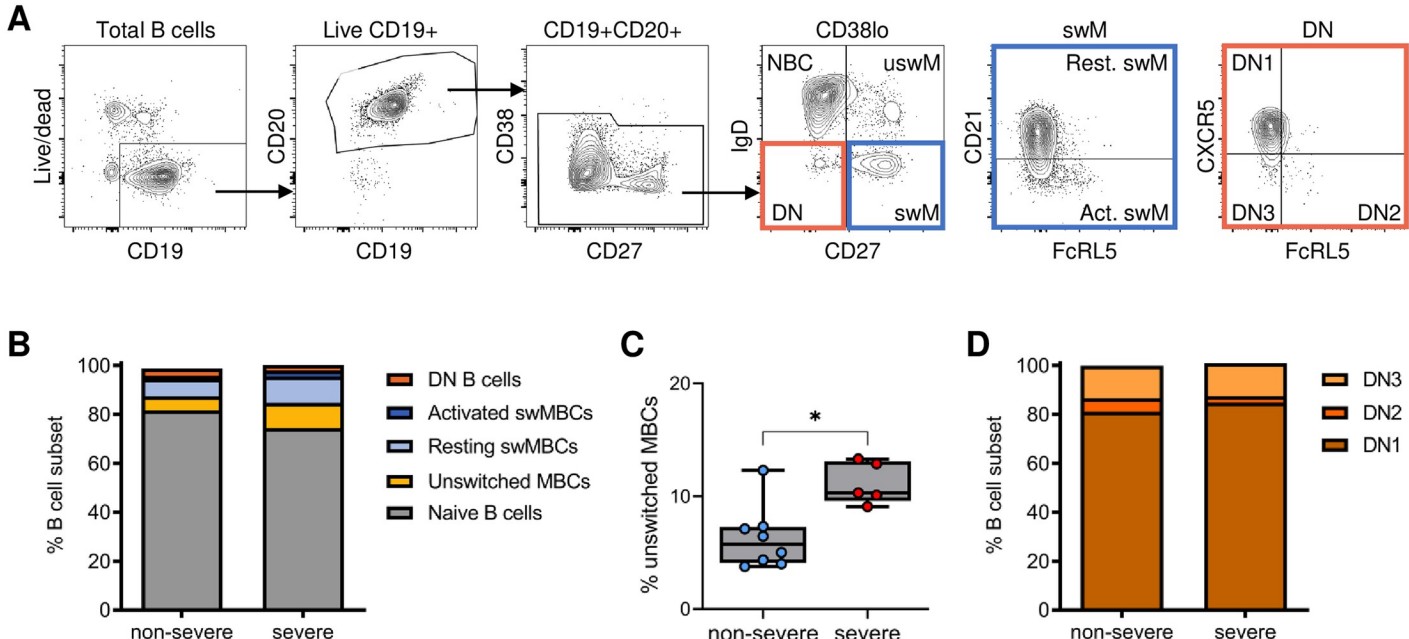

**Fig 2. Distribution of major B cell subsets in recovered COVID-19 patients.** A) Gating strategy to obtain non-antibody-secreting B cells (CD38lo) that are further divided into naïve, unswitched memory (uswM), switched memory (swM), and double negative (DN). SwM B cells were divided into activated and resting populations based on the expression of CD21. DN cells were divided into DN1–3 based on the expression of FcRL5 and CXCR5. B) Median percentage of each B cell subset in samples from individuals who recovered from non-severe COVID-19 and severe COVID-19. C) Percentage of unswitched memory B cells, which was increased in individuals who recovered from severe disease as compared to those who had non-severe COVID-19. D) Median proportions of the three different DN populations 1 month after non-severe or severe disease. In panel B–D, results are shown for individuals who recovered from non-severe COVID-19 (n = 8) and severe COVID-19 (n = 5). See S1 Fig for graphs with individual data points for the data shown in panels B and D. * P < 0.05.

## The percentage of spike-specific and RBD-specific B cells and their isotype distributions are similar between non-severe and severe cases

Our flow cytometry panel also included antigen probes to detect B cells reactive with the SARS-CoV-2 spike protein and the spike receptor binding domain (RBD). RBD is the part of the spike protein that interacts with angiotensin-converting enzyme 2, the viral receptor on host cells, and is the dominant target of neutralizing antibodies [34]. However, the majority of antibodies generated against the spike protein bind to epitopes outside RBD [35]. Whereas the RBD sequence is highly specific for SARS-CoV-2 [36], the non-RBD parts of spike, in particular the S2 subunit, share epitopes with other coronaviruses that circulate in the human population [37–39]. Evidence for pre-existing memory B cells that are cross-reactive with other coronaviruses has been reported [40]. Because pre-existing immunity could affect the development of the immune responses against these two parts of the spike protein, we analyzed B cell responses against RBD and non-RBD spike epitopes separately.

We constructed spike tetramers in two fluorochrome formats and defined the population of cells staining positive for both tetramers as spike-specific (Fig 3A, S2 Fig). The spike-specific B cells were then divided into non-RBD-specific and RBD-specific B cells based on reactivity with the RBD tetramer (Fig 3A). Non-RBD-specific and RBD-specific B cells were strongly enriched for antigen-experienced B cells (Fig 3B, compare to Fig 2B), with no statistically significant differences in the proportions of various B cell populations between individuals who recovered from non-severe or severe disease (Fig 3B, S3 Fig). In addition, reactivity against

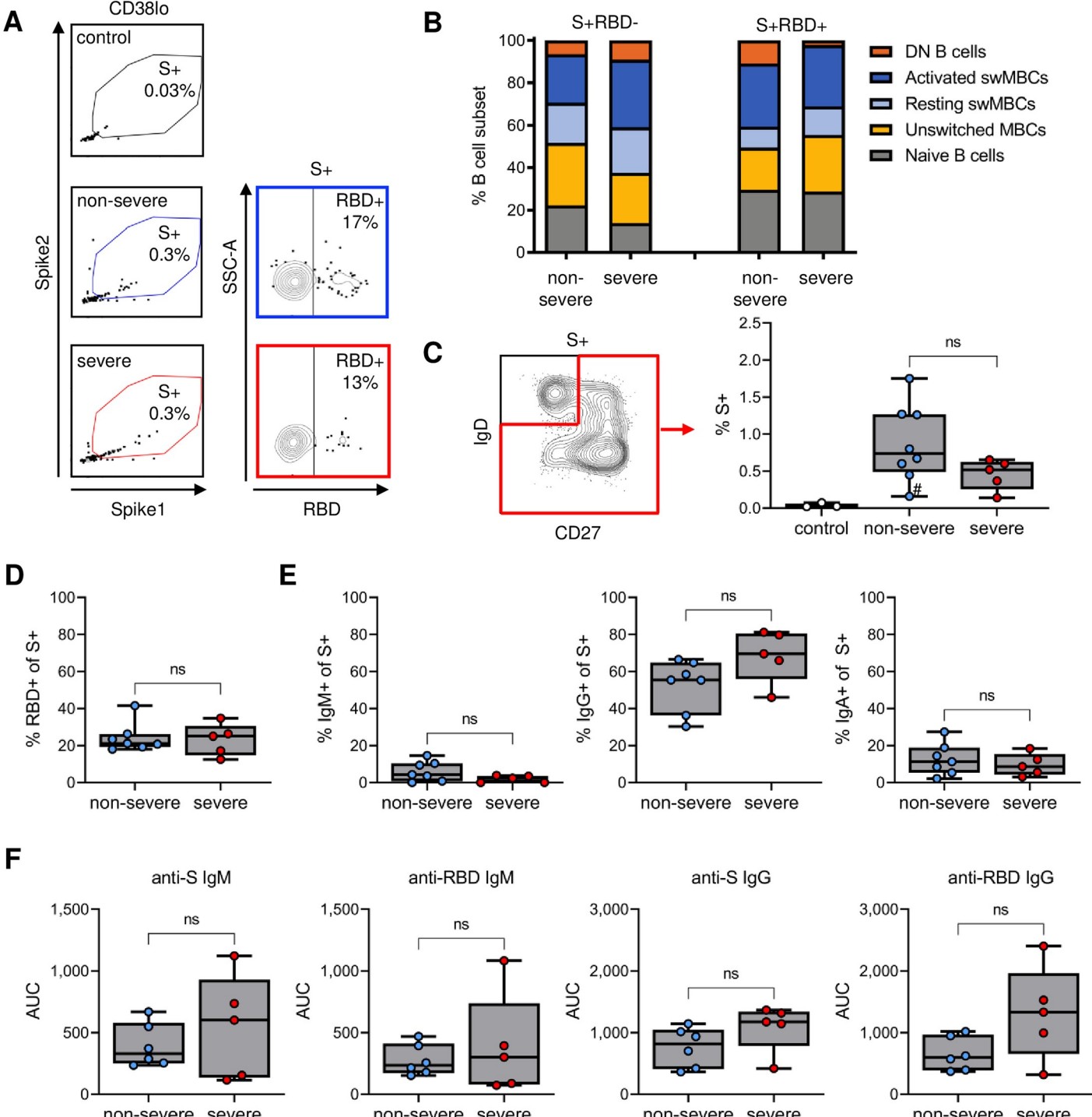

**Fig 3. Detection of spike-specific and RBD-specific B cells and plasma antibody responses.** A) Flow cytometry gating of spike-specific (S+) and RBD-specific (RBD +) B cells. B) Phenotype of non-RBD-specific (S+RBD-) and RBD-specific (S+RBD+) B cells, showing the median percentage among each group. Graphs with individual data points for each B cell subset are presented in S2 Fig. C) Percentage of antigen-experienced spike-specific B cells in individuals who have never been exposed to SARS-CoV-2 (control) or who recovered from non-severe or severe disease. D) Percentage of spike-specific B cells that recognized RBD. E) Percentage of class-switched spike-specific B cells that express IgM (left), IgG (middle), or IgA (right). F) Plasma IgM (left) and IgG (right) titers against spike and RBD. Note that the range of the Y axes are different between IgM and IgG. In panel C, results are shown for individuals who recovered from non-severe COVID-19 (n = 8) and severe COVID-19 (n = 5). Only seven individuals with non-severe COVID-19 were included in panels B, D, and E, due to the low number of spike-specific B cells detected in the individual marked with # in panel C.

spike and RBD tetramers among control donor B cells was minimal (Fig 3A and 3C), suggesting that these probes selectively bind to antigen-specific B cells.

The percentage of spike-specific B cells among antigen-experienced B cell populations (unswitched memory, switched memory, and DN B cells) among all patients ranged from 0.1% to 1.8% (median, 0.6%) (Fig 3C). One donor (marked with # in Fig 3C) had only six spike-specific B cells and was therefore not included in subsequent analyses that required further subsetting of these cells. All other donors had between 42 and 450 spike-specific B cells (median n = 121). In line with a previous study [4], a median of 22% of spike-specific B cells (median n = 18, range 7–84) were reactive against RBD (Fig 3D). Ogega *et al.* recently reported that patients who recovered from severe COVID-19 harbored more RBD-specific B cells than COVID-19 patients who never required hospitalization [29]. We did not observe a difference in the percentage of RBD-specific cells between individuals who recovered from non-severe and severe disease, potentially because our non-severe group had been hospitalized and suffered from more severe disease than the group with non-severe disease in the Ogega *et al.* study.

Next, we compared the isotype of class-switched spike-specific B cells between patients who recovered from non-severe and severe COVID-19 by categorizing spike-specific antigen-experienced B cells based on IgM, IgG, or IgA expression. The percentage of IgM$^+$, IgG$^+$, and IgA$^+$ class-switched B cells among spike-specific B cells did not differ significantly between the two patient groups and in both groups, the majority of class-switched spike-specific B cells were IgG$^+$ (Fig 3E). We also determined plasma IgM and IgG reactivity to the spike protein and RBD and observed no significant differences in IgM or IgG titers to the spike protein or RBD between convalescent patients who recovered from non-severe or severe disease (Fig 3F). Collectively, these results suggest that individuals who recovered from either non-severe or severe COVID-19 have similar immune responses to the SARS-CoV-2 spike protein, in terms of the prevalence of spike-specific B cells and their major phenotype.

## Non-severe COVID-19 is associated with an increased population of T-bet + spike-specific IgG+ B cells

To gain deeper understanding of the differences in memory B cell responses between individuals who had either non-severe or severe COVID-19, we combined the acquired flow cytometry data for all 19 markers (S2 Table) of all individuals and plotted a composite image using Uniform Manifold Approximation Projection (UMAP). UMAP clusters cells in a 2D plot based on similarity in phenotype and therefore provides meaningful organization of cell subsets. For each recovered COVID-19 patient, we took a random sample of cells (n = 10,000) and projected the spike-specific antigen-experienced B cells onto the composite UMAP (Fig 4A). We also plotted contours for various B cell subsets to visualize their location within the UMAP (Fig 4B, see S4 Fig for heatmaps of all individual markers overlaid on the composite UMAP). The composite image shows differences in the location of spike-specific B cells, predominantly among IgG$^+$ B cells (Fig 4A and 4B). The large majority of IgG$^+$ B cells are memory B cells, but a small fraction of these cells are DN1 B cells that are thought to contribute to a functional immune response and were therefore included in a phenotypic analysis of spike-specific IgG$^+$ B cells.

Irrespective of disease severity, spike-specific IgG$^+$ B cells expressed increased levels of activation markers CD80, Ki-67, and CD95 compared to all IgG$^+$ B cells, with no differences between the two groups (Fig 4C, S5 Fig). In contrast, spike-specific IgG$^+$ B cells after non-severe disease showed an increase in the expression of the transcription factor T-bet that was not seen after severe disease (Fig 4D). A median of 28% of spike-specific IgG$^+$ B cells in

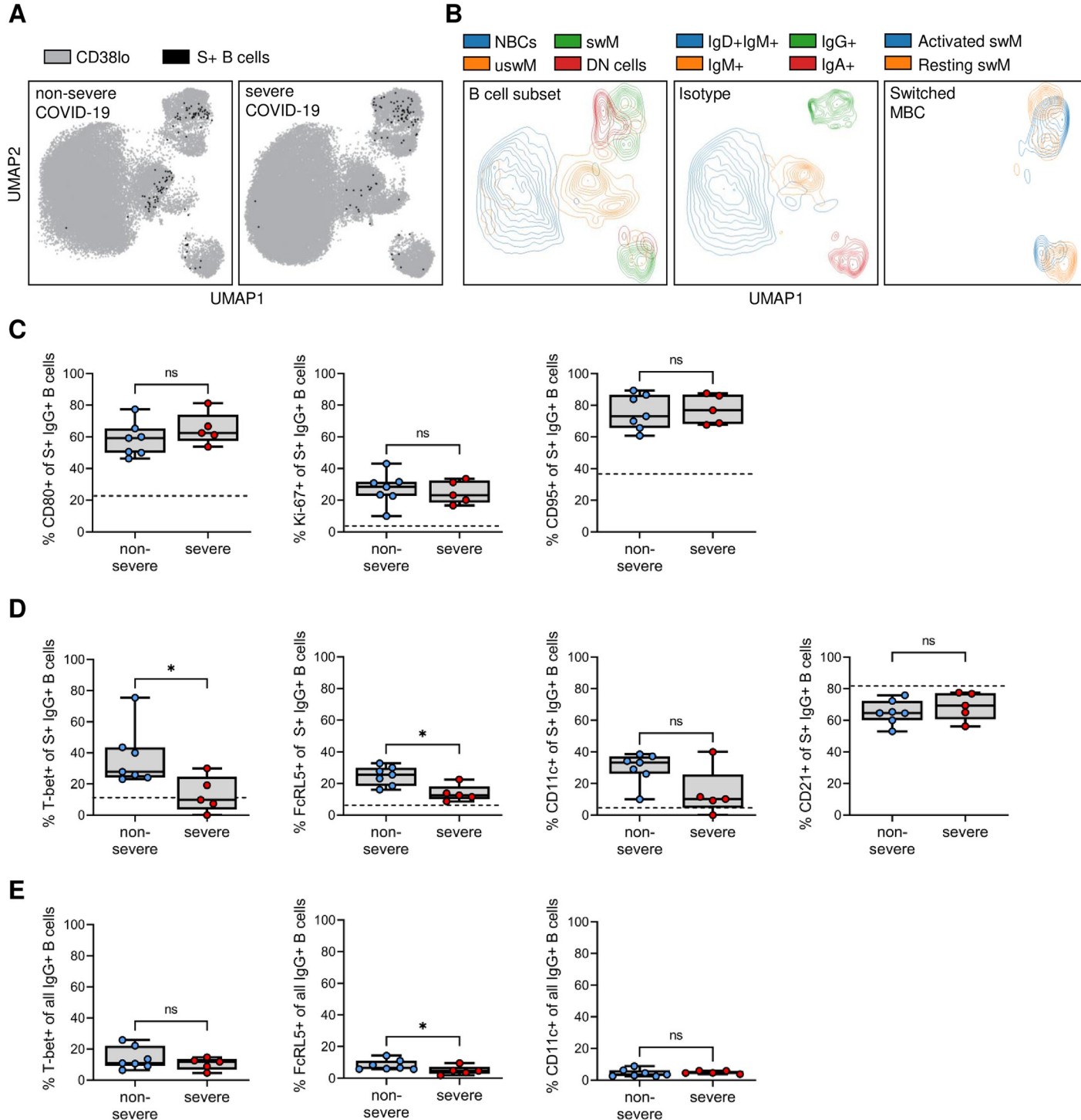

**Fig 4. Differences in the percentage spike-specific T-bet⁺ and FcRL5⁺ B cells between patients recovered from non-severe and severe COVID-19.** A) Composite UMAP showing the overlay of spike-specific B cells from individuals who recovered from non-severe (left) or severe (right) disease onto all B cells from that group. B) Overlay of major B cell subsets and isotypes onto the composite UMAP. C) Expression of CD80, Ki-67, and CD95 by spike-specific (S+) IgG⁺ B cells in individuals who experienced non-severe (n = 7) or severe (n = 5) COVID-19. D) Expression of T-bet, FcRL5, CD11c, and CD21 by spike-specific IgG⁺ B cells in individuals who experienced non-severe or severe COVID-19. In all plots, the median expression level of the marker of interest in all IgG⁺ B cells from both non-severe and severe COVID-19 recovered patients are indicated with a dashed line (see also S5 Fig). E) Expression of T-bet, FcRL5, and CD11c by all IgG⁺ B cells in individuals who experienced non-severe or severe COVID-19. * P < 0.05.

individuals who experienced non-severe disease expressed T-bet, which has been associated with strong anti-viral immunity [41]. Conversely, T-bet was expressed in only about 10% of spike-specific IgG+ memory B cells in individuals who recovered from severe disease (Fig 4D, S6 Fig). T-bet+ cells often have a distinct cell surface marker profile, including expression of CD11c and FcRL5 and the absence of CD21 [25, 26, 28]. Indeed, in addition to T-bet, the percentage of FcRL5+ spike-specific IgG+ memory B cells was increased after non-severe disease as compared to severe disease (Fig 4D). The expression of CD11c and CD21 was not statistically significantly different between the two groups (Fig 4D). However, the percentage of CD11c+ spike-specific IgG+ memory B cells was increased among spike-specific IgG+ B cells in patients who recovered from non-severe COVID-19 as compared to all IgG+ B cells in these patients, while this difference was not seen in individuals who recovered from severe COVID-19 (S5 Fig). Conversely, a lower percentage of CD21+ spike-specific IgG+ B cells was observed as compared to all IgG+ B cells after non-severe disease but not after severe disease (S5 Fig). Additionally, when comparing expression profiles of T-bet, FcRL5, and CD11c on total IgG+ B cells, we only observed a small difference between the two groups for FcRL5 (Fig 4E). However, the relative increase in FcRL5 expression was higher in spike-specific B cells (2.0-fold) than in total IgG+ B cells (1.4-fold), suggesting that differences between spike-specific B cells are mainly driven by the response to antigen, not by a more generalized effect of infection on the B cell compartment. To confirm that the measurement of T-bet, FcRL5, CD11c, and CD21 expression was robust between experiments, we processed and analyzed technical replicates for samples from one non-severe and one severe case on different days, and observed almost perfect correlation between the replicates (Spearman r = 0.99, P = 0.0002, S7 Fig). We then analyzed the combinatorial expression of these markers by combining spike-specific IgG+ T-bet+ B cells from all donors and plotting contour plots depicting the expression of FcRL5, CD11c, CD21, CD27, CXCR3, and CXCR5 (Fig 5). This revealed four subpopulations of T-bet+ IgG+ B cells, of which cells expressing FcRL5, CD11c, CD27 and CXCR3, and lacking CD21 and CXCR5 (type I; activated switched memory B cells) were most abundant in both non-severe and severe cases (Fig 5). This phenotype has previously been observed in activated

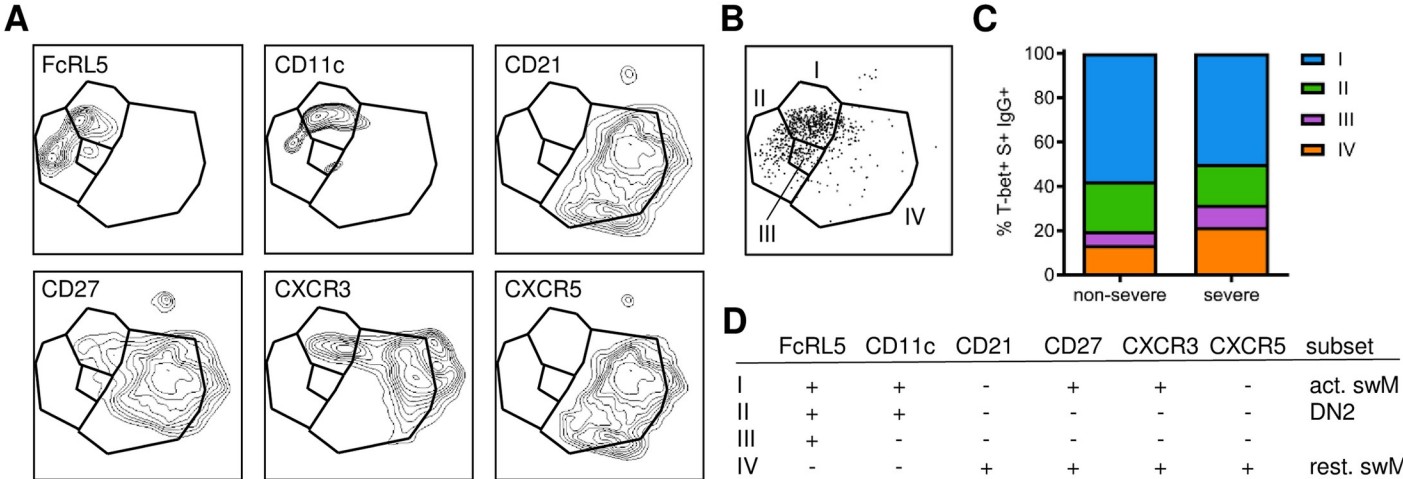

**Fig 5. The phenotype of T-bet+ IgG+ spike-specific B cells.** A) UMAPS showing the expression of individual markers in T-bet+ IgG+ spike-specific B cells. B) Gating of four populations of T-bet+ IgG+ B cells based on unique expression profiles of the markers shown in panel A. C) Distribution of the four T-bet+ IgG+ B cell population among individuals who recovered from non-severe (n = 7) and severe (n = 5) COVID-19. D) Summary of the phenotype of the four T-bet+ IgG+ B cell populations identified here and their classification into one of the major B cell subsets: activated switched memory (act. swM), DN2, or resting switched memory (rest. swM) B cells.

B cells after influenza virus vaccination and is thought to delineate a B cell subset associated with long-lived humoral immunity [26, 28, 41]. Collectively, the higher prevalence of spike-specific B cells that express markers associated with durable immunity in individuals who recovered from non-severe COVID-19 than in those who experienced severe disease suggests that disease severity influences the quality of the B cell response.

### SARS-CoV-2 spike-specific memory B cells return to a resting phenotype with baseline levels of T-bet expression five months post-symptom onset

Activated T-bet[+] B cell subsets have been shown to expand and contract rapidly following influenza virus vaccination, peaking between 14 and 28 days post-vaccination and returning to baseline levels after three months [25]. In addition, it has been shown that the memory B cell response continues to mature up to one year following SARS-CoV-2 infection, giving rise to B cell clones with higher levels of somatic hypermutation and resulting in increases in antigen-binding affinity and ability to neutralize the virus [7, 19, 20]. To evaluate the early dynamics of T-bet[+] spike-specific B cells and continued development of the phenotype of memory B cells in parallel with ongoing selection and maturation, we analyzed additional blood samples from recovered COVID-19 patients with non-severe disease collected at an earlier time point (median, 19 days post-symptom onset) (T1, Fig 1) and at a follow-up visit five months post-symptom onset (T3, Fig 1). Unfortunately, we were unable to recruit any individuals who had experienced severe disease for a follow-up visit at five months post-symptom onset. In addition, at the earlier time point, most individuals with severe disease were still hospitalized (Fig 1). We therefore limited this analysis to individuals who recovered from non-severe COVID-19.

The percentages of total naïve, unswitched memory, resting switched memory, activated switched memory, and double negative B cell subsets among total B cells did not differ between the two time points (S8 Fig). However, the phenotype of spike-specific B cells changed considerably between the two early time points and five months post-symptom onset (Fig 6A, S9 Fig), as described in further detail below. Similar to what has been reported by Dan *et al.* and Sokal *et al.* [4, 8], the percentage of spike-specific B cells among antigen-experienced B cells remained relatively stable at five months post-symptom onset (Fig 6B). In addition, the reactivity against RBD among these spike-specific B cells increased between 2–3 weeks and 4–5 weeks post-symptom onset, but did not change at the 5-month time point (Fig 6C). The larger proportion of non-RBD-specific B cells early in infection may be the result of recall responses of pre-existing memory B cells that are cross-reactive with other coronaviruses.

The percentage of antigen-experienced spike-specific B cells with an unswitched memory B cell phenotype was reduced at the 5-month time point (Fig 6D). The percentage of spike-specific activated switched memory B cells drastically decreased over time, accompanied by an increase in resting memory B cells (Fig 6E). Among switched memory B cells, the percentage of IgM[+] and IgA[+] memory B cells declined and the proportion of IgG[+] memory B cells increased at five months post-symptom onset (Fig 6F). In line with the decrease of IgM[+] memory B cells, the anti-spike and anti-RBD plasma IgM responses were lost almost completely at five months post-symptom onset. Plasma IgG titers against the spike protein remained stable, whereas plasma IgG titers against RBD showed a modest decrease (Fig 6G). These results indicate that both the unswitched plasma cell and memory B cell response against the spike protein are short-lived, while the class-switched IgG[+] response is more durable, as has been reported by others [4, 8].

Both the decrease of unswitched memory B cells and the change from an activated to a resting switched memory B cell phenotype can be seen in composite UMAPs of the three time

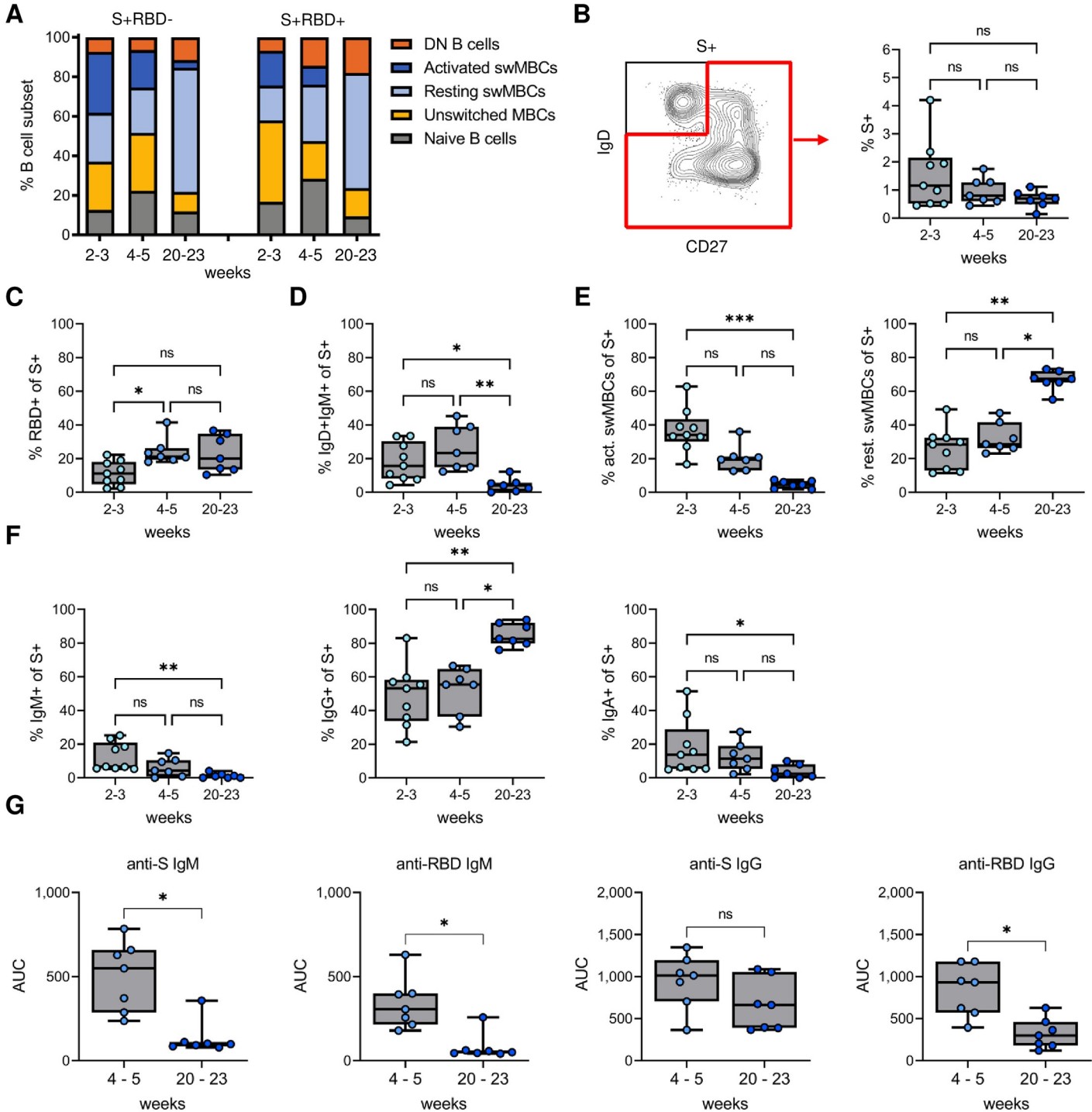

**Fig 6. Change from activated to resting memory phenotype of IgG+ spike-specific B cells at five months post-symptom onset.** A) Phenotype distribution of non-RBD-specific (S+RBD-) and RBD-specific (S+RBD+) B cells, showing the median percentage among each group. Graphs with individual data points for each B cell subset are presented in S6 Fig. B) Percentage of antigen-experienced spike-specific B cells at 2–3, 4–5, and 20–23 weeks post-symptom onset. C) Percentage of S+RBD + B cells among antigen-experienced spike-specific B cells. D) Percentage of unswitched memory B cells among antigen-experienced spike-specific B cells. E) Percentage of activated (left) and resting (right) switched memory B cells among antigen-experienced spike-specific B cells. F) Percentage of class-switched spike-specific B cells that express IgM (left), IgA (middle), or IgG (right). G) Plasma IgM (left) and IgG (right) titers against spike and RBD. In all panels, results are shown for samples collected 2–3 (n = 9), 4–5 (n = 7), and 20–23 (n = 7) weeks post-symptom onset. * P < 0.05; ** P < 0.01; *** P < 0.001.

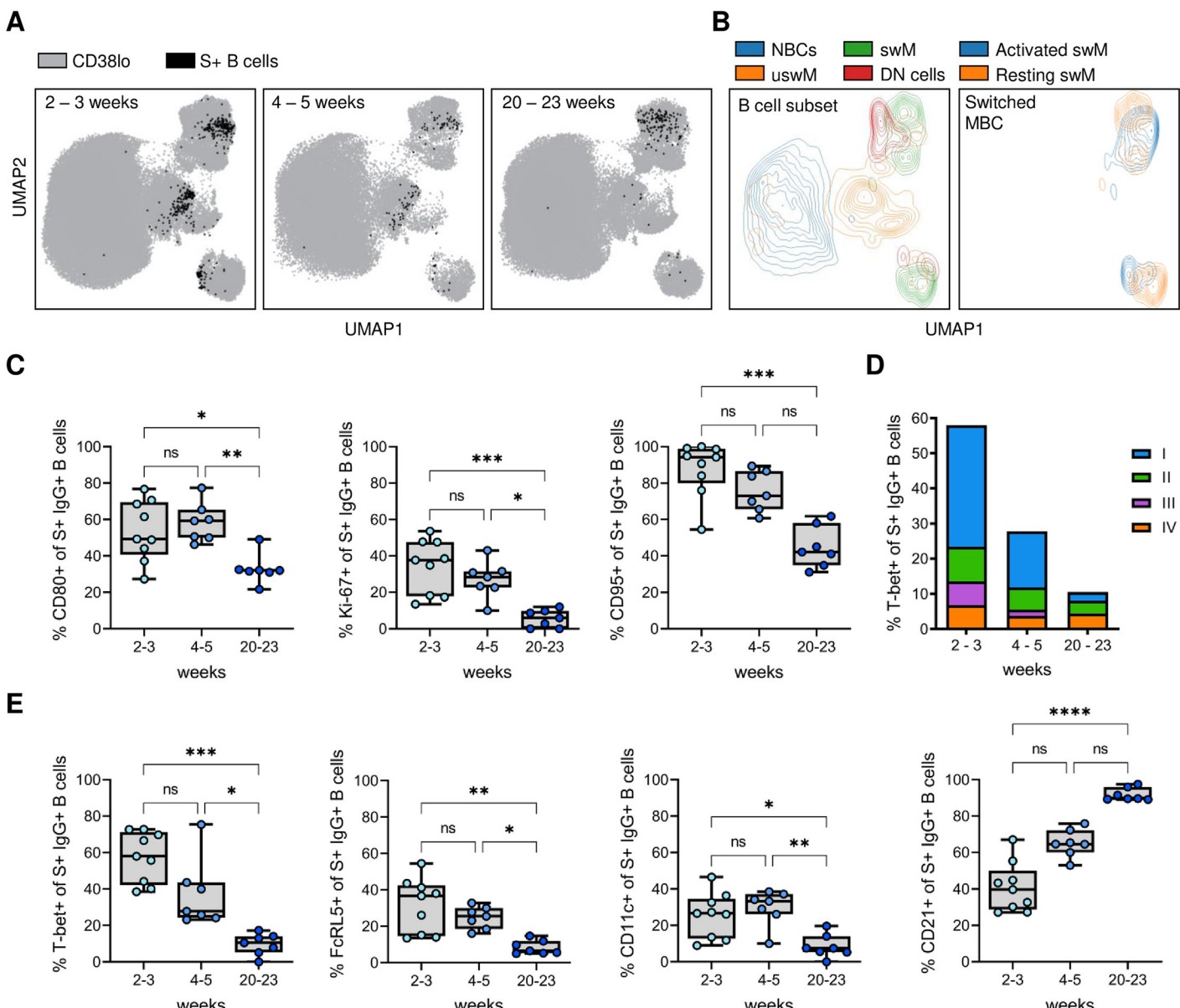

**Fig 7. Longitudinal dynamics in the percentage of spike-specific B cells that express T-bet and various surface markers shortly after recovery and five months post-symptom onset.** A) Composite UMAP showing the overlay of spike-specific B cells from individuals who recovered from non-severe COVID-19 at 2–3 (left), 4–5 (middle), and 20–23 (left) weeks post-symptom onset. B) Overlay of major B cell subsets and activated/resting switched memory B cells onto the composite UMAP. C) Expression of CD80, Ki-67, and CD95 by spike-specific (S+) IgG⁺ B cells at the three time points. D) Distribution of the four T-bet⁺ subsets (as defined in Fig 5) among spike-specific IgG⁺ B cells at the three time points post-symptom onset. E) Expression of T-bet, FcRL5, CD11c, and CD21 by spike-specific IgG⁺ B cells at the three time points post-symptom onset. In all graphs, results are shown for samples collected 2–3 (n = 9), 4–5 (n = 7), and 20–23 (n = 7) weeks post-symptom onset. $^*$ P < 0.05; $^{**}$ P < 0.01; $^{***}$ P < 0.001; $^{****}$ P < 0.0001.

points (Fig 7A and 7B). In line with the return to a resting memory B cell phenotype, expression of CD80, Ki-67, and CD95 was decreased at 20–23 weeks post-symptom onset as compared the two earlier time points (Fig 7C). The percentage of CD80⁺ spike-specific IgG⁺ B cells at 20–23 weeks post-symptom onset still showed an increase over baseline levels (P = 0.016, Wilcoxon signed rank test), while expression of Ki-67 and CD95 had returned to levels seen in non-spike-specific IgG⁺ B cells (Fig 7C, S10 Fig). The expression of T-bet was higher at 2–3

weeks post-symptom onset than at 4–5 weeks post-symptom onset (median, 58% and 28%, respectively), although this difference was not statistically significant, and decreased further to baseline levels (~ 10%) at 20–23 weeks post-symptom onset (Fig 7D and 7E). Similar dynamics were observed for FcRL5 and CD11c, while CD21 expression increased significantly over time (Fig 7E). The percentage of type I T-bet$^+$ spike-specific IgG$^+$ B cells (activated switched memory B cells) that dominated the early response decreased dramatically over time (Fig 7D). Instead, the T-bet$^+$ spike-specific IgG$^+$ B cells present at five months post-symptom onset were mainly type II (FcRL5$^+$ CD11c$^+$ CD21$^-$ CD27$^-$, DN2 B cells) and type IV (FcRL5$^-$ CD11c$^-$ CD21$^+$ CD27$^+$, resting switched memory B cells) subsets, making up ~35% and ~40% of T-bet$^+$ spike-specific IgG$^+$ B cells, respectively. These results suggest that the memory B cell response continues to evolve until at least five months after infection. In addition, the decrease of T-bet$^+$ spike-specific B cells in the circulation over the course of five months post-symptom onset suggests that their abundant presence in the blood may signify recent exposure.

## Discussion

In the current COVID-19 pandemic, an important question that has only partially been answered is whether SARS-CoV-2 infection elicits durable immunity. Assessments of B cell responses up to one year post-infection suggest that both neutralizing antibodies and memory B cells against the SARS-CoV-2 spike protein remain detectable in the circulation of most recovered COVID-19 patients and are stable or decay slowly at this time point. In this study, we aimed to better understand the memory B cell response and to determine whether the severity of COVID-19 disease course influences the development of B cell responses. Using high-parameter spectral flow cytometry, we analyzed the phenotype of B cells reactive with both non-RBD and RBD epitopes on the spike protein shortly after recovery and approximately five months post-symptom onset.

Our study consisted of a relatively small number of individuals but recapitulated many of the observations reported from much larger cohorts, including the frequency of spike-specific memory B cells after infection and the fraction of spike-specific B cells that recognizes RBD epitopes. We also confirmed the loss of IgM$^+$ and IgA$^+$ spike-specific memory B cells and anti-spike and anti-RBD plasma IgM, as well as maintenance of IgG$^+$ spike-specific memory B cells and anti-spike plasma IgG five months post-symptom onset [4, 17]. This gives us confidence in the validity of several new observations that we discuss in more detail below. While antigen-specific B cells make up only a small percentage of total B cells (<1%), we detected a meaningful population of spike-specific B cells (between 42 and 450 cells) in all except one donor, who was therefore excluded from further analysis. RBD-specific B cells were sparser (between 7 and 84 cells per donor) and all main observations are therefore based on the larger populations of spike-specific B cells.

The most striking result from this study is the higher percentage of spike-specific IgG$^+$ B cells that express the transcription factor T-bet in individuals who recovered from non-severe disease as compared to those who recovered from severe disease. Previously, it was observed that a higher percentage of total CD19$^+$ T-bet$^+$ IgG$_1$$^+$ B cells was associated with shorter symptom duration [42]. Here, we determined that this association predominantly involves SARS-CoV-2 antigen-specific B cells, suggesting implications for the development of B cell memory. The expression of T-bet, in parallel with a surface marker profile that includes the presence of FcRL5 and CD11c and the absence of CD21, has previously been reported on activated B cells shortly after vaccination and infection, and is associated with long-lived humoral immunity [26, 28, 41]. Here, we observed an increase in the percentage of spike-specific IgG$^+$ B cells that expressed T-bet and FcRL5 shortly after recovery from non-severe COVID-19, but not after

severe COVID-19. This increased expression was only observed in SARS-CoV-2 spike-specific B cells for T-bet and was larger in spike-specific B cells as compared to total B cells for FcRL5, suggesting an alteration exclusively in the B cell response to the virus and not a global change to the B cell compartment as a result of infection. Ogega *et al.* reported higher FcRL5 expression on RBD-specific class-switched memory B cells after non-severe COVID-19 (non-hospitalized patients) than after severe disease (hospitalized patients) [29], findings that are in agreement with our results. In addition, higher levels of somatic hypermutation were observed in IgG[+] memory B cells following non-severe COVID-19 than severe disease [43], while patients who died from COVID-19 showed an absence of germinal centers and a delay in the development of neutralizing antibodies [24, 44]. Collectively, these data are indicative of functional development of B cell immunity with germinal center responses in patients with non-severe disease, as compared to a more dysfunctional B cell response in severe disease cases.

T-bet expression in B cells can be induced by interferon gamma (IFNγ) signaling through the IFNγ receptor [45, 46]. IFNγ plays an important role in the immune response against viral infections. Although many contradictory findings have been reported about the role of INFγ in SARS-CoV-2 infection, the induction of INFγ early in infection seems important in the control of viral replication (summarized in [47]). Indeed, low IFNγ and low induction of interferon-stimulated genes contribute to the development of COVID-19 [48]. Failure to control viral load leads to the induction of a hyper-inflammatory response later in disease, in parallel with persistent high levels of IFNγ. However, these conditions may not promote the development of T-bet[+] memory B cells. Instead, patients with severe disease show a strong extrafollicular B cell response, resulting in an accumulation of DN2 B cells [22]. In our study, no difference was observed in the percentage of DN2 B cells shortly after recovery between patients with either non-severe or severe disease, suggesting that these DN2 B cells may be relatively short-lived. Of note, DN2 cells typically also express T-bet [21], highlighting that the expression of this transcription factor alone is not the determining factor in the development of either a functional or pathogenic B cell response. The difference that we observed in the percentage of T-bet[+] spike-specific IgG[+] B cells between patients recovered from non-severe or severe COVID-19 could reflect these two developmental pathways of anti-viral immune responses.

In individuals who experienced non-severe COVID-19, T-bet[+] spike-specific IgG[+] B cells almost disappeared from the circulation five months post-symptom onset. This is in line with the transient peak of activated memory B cells observed after influenza vaccination [25, 26]. The cell surface marker expression profile and dynamics of the T-bet[+] B cells detected in this study matches best with the AM2 subset of activated memory B cells defined by Andrews *et al.*, which peaked 14 days after influenza vaccination and had almost disappeared 90 days post-vaccination [25]. However, this does not necessarily mean that these cells are short-lived. It was recently shown that the spleen harbors a population of resident T-bet[hi] memory B cells with specificity to viral antigens [49]. In addition, it has been reported that T-bet[hi] memory B cells elicited in response to influenza virus vaccination rapidly differentiated into antibody-secreting cells upon reactivation one year later [28]. T-bet[+] memory B cells thus seem to contribute to effective recall responses upon re-exposure, but may only be detectable in the circulation shortly after infection. In this regard, it is important to point out that the timing of sampling can strongly affect the frequency of transient B cell populations, and samples selected for comparative analyses need to be carefully matched to account for rapid changes in the abundance of activated B cell populations over time. For example, this may explain the discrepancy between this study and the observation by Sokal *et al.* that T-bet expression in spike-specific B cells was higher in individuals with severe COVID-19 than individual with non-severe COVID-19 [8]. In the Sokal *et al.* study, samples from severe cases were collected at an

earlier time point than those from non-severe cases (median, 18.8 days versus 35.5 days post-symptom onset), which could have influenced these results [8].

Thus far, no differences have been observed in serum anti-spike antibody titers, *in vitro* serum neutralizing activity, or percentages of spike-specific memory B cells between non-severe and severe COVID-19 patients in the months after infection [18]. It remains to be determined whether the durability of B cell memory differs between individuals who experienced non-severe or severe disease. We speculate that individuals who recovered from non-severe disease develop a higher quality memory B cell response that may be longer-lived or may give rise to a larger population of antibody-secreting cells upon re-exposure. An alternative interpretation of the paucity of T-bet[+] spike-specific memory B cells in patients with severe disease could be that a robust T-bet[+] B cell response protects against severe COVID-19. Although patients who died from COVID-19 show defects in the development of B cell responses [24, 44], it is likely that the absence of a population of T-bet[+] spike-specific memory B cells is a consequence of the immune environment during disease, not a factor contributing to pathogenesis. We believe these cells are a sign of a balanced immune response that contributes to viral clearance through the production of neutralizing antibodies and leads to the development of durable B cell memory, but that T-bet[+] memory B cells do not play a direct role in protection against severe disease.

In conclusion, we have shown that the memory B cell response against SARS-CoV-2 spike protein develops differently in patients with non-severe disease compared those with severe disease, with more spike-specific B cells that express a B-cell marker profile associated with durable immunity, characterized by expression of T-bet and FcRL5 in individuals who recovered from non-severe COVID-19. Although antibody titers or percentage of spike-specific memory B cells up to one year are similar or higher in individuals who recovered from severe disease as compared to non-severe cases [8, 18], the increased percentage of B cells associated with long-lived immunity in non-severe COVID-19 patients may have consequences for long-term immunity against SARS-CoV-2 re-infection or severity of the resulting disease. T-bet[+] spike-specific B cells nearly disappeared from the circulation five months post-symptom onset, consistent with loss of these cells from the periphery or migration into tissues. These data aid in the understanding of naturally acquired B cell responses against SARS-CoV-2 and help characterize the B cell populations that may be responsible for durable, long-lived immunity.

# Materials and methods

## Isolation of plasma and peripheral blood mononuclear cells

Blood from recovered COVID-19 patients was processed within four hours of blood draw. Plasma was separated from cells by centrifugation at $250 \times g$ for 5 min at room temperature (RT). Plasma was depleted of platelets by centrifugation at $2,000 \times g$ for 15 min at 4˚C and was stored at -20˚C. Peripheral blood mononuclear cells (PBMCs) were isolated from the cellular fraction as described below. PBMCs used as negative controls were collected in 2018 and early 2019 by isolation from buffy coats (Interstate Blood Bank, Memphis, TN) obtained the day following blood draw. Buffy coats were stored and shipped at room temperature (RT) and processed immediately upon arrival. Cells were diluted approximately $4 \times$ in PBS with 2 mM EDTA. PBMCs were layered on Ficoll-Paque (GE Healthcare, cat. no. 17144002) and spun at $760 \times g$ for 20 min at RT to pellet erythrocytes and separate leukocytes. The leukocytes were then washed in PBS with 2 mM EDTA, centrifuged at $425 \times g$ for 15 min at RT, followed by centrifugation at $250 \times g$ for 10 min at RT for all subsequent wash steps necessary to remove all platelets, determined by cloudiness of the supernatant. Next, PBMCs were resuspended in cold IMDM/GlutaMAX (Gibco, cat. no. 31980030) supplemented with 10% heat-inactivated,

USA-sourced, fetal bovine serum (FBS) and 5% cell culture grade DMSO, counted, and cryo-preserved in the liquid nitrogen vapor phase.

## Spike and RBD protein production and tetramer synthesis

Recombinant SARS-CoV-2 HexaPro spike and RBD were produced from Addgene plasmid #154754 [50] and BEI plasmid NR-52309, respectively, for use in ELISA experiments. DNA plasmids were purified from DH5alpha Competent Cell (Zymo, cat. no. T3007) cultures using a ZymoPURE II Plasmid Maxiprep (Zymo, cat. no. D4203). 25 μg plasmid was used to trans-fect 25 ml Expi293F (Thermo, cat. no. A14635) culture per manufacturer's instructions. Cul-ture supernatants were collected 5 days post-transfection. His-tagged S and RBD were purified using HiTrap chelating high performance columns (Cytiva, cat. no. 17-0408-01) charged with $Ni^{2+}$, washed with PBS (pH 7.2), and eluted with 0.3 M imidazole in PBS (pH 7.2). The imidaz-ole was removed from the eluted protein buffer using a HiPrep desalting column (Cytiva, cat. no. 17-5087-01). The protein was concentrated using a 30 kDa MWCO Amicon ultra-15 cen-trifugal filter (Sigma, cat. no. UFC903024), and stored at -70°C. For the production of biotiny-lated S and RBD protein used for the synthesis of antigen tetramers, Addgene plasmid #154754 and BEI plasmid NR-52309 were altered to include a biotinylation site. The modified plasmids have been deposited to Addgene: #166856 and #166857 for spike and RBD, respec-tively. Cells were co-transfected with either plasmid #166856 or #166857 and a plasmid encod-ing BirA ligase (Addgene #32408) at a 4:1 ratio (m/m). 100 mM D-biotin (Sigma, cat. no. B4639) was added immediately post-transfection to a final concentration of 10 mM. Spike and RBD tetramers were synthesized by incubating biotinylated protein with fluorophore-conju-gated streptavidin overnight at 4°C (streptavidin-PE, Tonbo, cat. no. 50-4317-U100, streptavi-din-APC, Tonbo, cat. no. 20-4317-U100, streptavidin-BV421, BioLegend, cat. no. 405226) at a molar ratio of 5:1, respectively, to generate three individual tetramers: spike-PE, spike-APC, and RBD-BV421.

## Protein gels

As a quality control step, purified spike and RBD proteins were visualized on a polyacrylamide gel (S11 Fig). 800 ng of protein was loaded on a 4–12% Bis-Tris gel (Thermo, cat. no. NP0321BOX) after it was mixed with 2× Laemmli loading buffer and incubated at 85°C for 5 min. The gel was run with MOPS running buffer (Thermo, cat. no. NP0001) at 200 V for 50 min after which the proteins were stained using Imperial Protein Stain (Thermo, cat. no. PI24615) per manufacturer's instructions.

## B cell isolation and staining

Cryopreserved PBMCs were thawed in a water bath set at 37°C and immediately mixed with pre-warmed thawing medium (IMDM/GlutaMAX supplemented with 10% heat inactivated FBS and 0.01% Universal Nuclease (Thermo, cat. no. 88700). After brief centrifugation (250 × g, 5 min) at RT, the cell pellet was resuspended in warm thawing medium and viable cells were counted using trypan blue on a Cellometer Mini (Nexcelom) automated cell counter. Next, the PBMCs were pelleted (250 × g, 5 min, RT), resuspended in isolation buffer (PBS supplemented with 2% heat inactivated FBS and 1 mM EDTA) at 50 million live cells/ml, and filtered through a 35 μm sterile filter cap (Corning, cat. no. 352235) to break apart any aggregated cells. B cells were isolated using the EasySep Human B Cell Isolation Kit (StemCell, cat. no. 17954) according to manufacturer's instructions. After washing with cold PBS (250 × g, 5 min, RT), the isolated B cells were incubated with 1 μl Zombie UV Fixable Viability kit (Biolegend, cat. no. 423107) per 1 ml cell suspension for 30 min on ice. Cells were

subsequently washed with cold PBS with 1% bovine serum albumin (BSA), followed by an incubation at 4˚C for 30 min with an antigen tetramer cocktail (S2 Table). Next, the cells were washed again with cold PBS with 1% BSA and incubated at 4˚C for 30 min with a B cell surface marker antibody cocktail (S2 Table), after which the cells were washed again with cold PBS with 1% BSA and resuspended in 1 ml Transcription Factor Fix/Perm Concentrate (Tonbo, part of cat. no. TNB-0607-KIT) diluted with 3 parts Transcription Factor Fix/Perm Diluent (Tonbo) and incubated at 4˚C for 1 hour. After the incubation, the cells were washed twice with 3 ml of 1× Flow Cytometry Perm Buffer (Tonbo) and resuspended in intracellular marker antibody cocktail (S2 Table) diluted in 1× Flow Cytometry Perm Buffer. After an incubation at 4˚C for 30 min., the cells were washed twice with 3 ml 1× Flow Cytometry Perm Buffer and once with 3 ml cold PBS + 1% BSA. The cells were then fixed by adding 4% formaldehyde at a 1:1 ratio to sample (v/v), washed with 3 ml cold PBS + 1% BSA, and resuspended to 20–30 million cells/ml in PBS with 1% BSA, and filtered into a FACS tube through a 35 μm sterile filter cap. Cells were analyzed by flow cytometry immediately following fixation.

## Flow cytometry analysis

B cells were analyzed on a Cytek Aurora spectral flow cytometer equipped with five lasers. SpectroFlo QC Beads (Cytek, cat. no. SKU N7-97355) were run prior to each experiment for performance tracking. Quality control and LJ tracking reports were used to ensure machine performance and settings between different runs were comparable. B cells isolated from control PBMCs collected pre-pandemic were used for the compensation of the live/dead stain and for the unstained control. UltraComp eBeads (Thermo, cat. no. 01-2222-41) were used for compensation of all other single stain fluorophores. FlowJo was used for gating and quantifying cell frequencies. The cytometry analysis software OMIQ was used for the integration and dimension reduction analysis. In short, Uniform Manifold Approximation and Projection (UMAPs) where created by first pre-gating single/live/CD19$^+$/CD20$^+$/CD38$^{lo}$ cells. Features used for UMAP projection included mean fluorescence intensity of staining markers, excluding live/dead, RBD, spike1, and spike2 with default parameters (neighbors = 15, metric = Euclidean, random seed = 9889) and included all COVID-19 samples used in this study for initial projection. Subsequently, files were subsampled to include an equal representation of cells from each donor and concatenated within each group. For the projection of B cell subsets, Ig isotypes, and antigen-specific B cells onto the UMAP, gates were manually set to identify populations of interest using two-dimensional displays, which were then overlaid onto the UMAP projection.

## Detection of spike protein and receptor binding domain using an enzyme-linked immunosorbent assay

96-well plates (Corning, cat. no. 07-200-721) were coated with 50 μl of SARS-CoV-2 spike or RBD at a concentration of 2 μg per ml in PBS and incubated overnight at 4ºC. An additional plate was coated with goat anti-human IgG (Sigma, cat. no. I2136-1ML) at 4 μg/ml and IgM (Sigma, cat. no. I1636-2ML) at 8 μg/ml, and served as a positive control. Plates were washed twice using a gentle stream of deionized water from a faucet and subsequently incubated with 200 μl blocking buffer (PBS with 0.01% Tween-20 (Fisher, cat. no. BP337-100) and 3% non-fat dry milk (SACO)) for 1 hour at RT. Three-fold serial dilutions of 20 μg/ml human IgG or IgM were prepared in dilution buffer (PBS with 0.01% Tween-20 and 1% non-fat dry milk) and were added to the IgG or IgM-coated and blocked plates in 100 μl total volume. Plasma samples were heat inactivated prior to handling by incubating at 56˚C for 30 min. Plasma samples were then diluted in dilution buffer starting at a 1:50 dilution, with an additional 7 two-fold serial dilutions. The diluted plasma samples, 27 from convalescent patients exposed to SARS-CoV-2

and 1 from a non-infected human control, were added to the plates in a total volume of 200 μl. After a two-hour incubation at RT, the plates were washed once with 150 μl of PBST by removing the samples and adding PBST by pipetting and subsequently washed 6 times using a gentle stream of deionized water from a faucet. Next, 100 μl anti-human IgG or IgM—horseradish peroxidase (HRP) conjugates (BioLegend), diluted 1:2500 and 1:5000 in dilution buffer, respectively, was added and the plates were incubated for 1 hour at RT. After four subsequent washes with deionized water 100 μl TMB (Thermo, cat. no. PI34024) was added and the plates were incubated in the dark at RT. The oxidation of TMB was stopped by adding 100 μl 0.18 M $H_2SO_4$ when the wells containing the most diluted IgG/IgM standard started to color. The absorbance was measured at 450 nm using a Synergy H4 Hybrid Plate Reader (BioTek). The average background signal of wells incubated without plasma was subtracted from the absorbance values for each sample. All plasma samples were measured in duplicate and the average absorbance reading was used to calculate the area under the curve using GraphPad 9.

### Statistics

Statistical analyses were performed in GraphPad 9. Differences between two groups were evaluated using a Mann-Whitney U test. Differences between three or more groups were tested using a Kruskal-Wallis test, followed by comparisons between selected pairs of groups using Dunn's post-hoc test, corrected for multiple comparisons. Non-parametric tests were used because the limited sample size did not allow for reliable evaluation of normal distribution and homoscedasticity.

### Study approval

Samples and associated clinical data used in this study were received de-identified from the University of Texas Health San Antonio COVID-19 Repository. This repository was reviewed and approved by the University of Texas Health Science Center at San Antonio Institutional Review Board. All study participants provided written informed consent prior to specimen collection for the repository to include collection of associated clinical information and use of left-over clinical specimens for research. The COVID-19 Repository utilizes an honest broker system to maintain participant confidentiality and release of de-identified data or specimens to recipient investigators.

### Supporting information

**S1 Fig. Distribution of major B cell subsets in patients who recovered from non-severe or severe COVID-19.** A) The percentage of naïve B cells (NBC; IgD$^+$ CD27$^-$), unswitched memory B cells (MBCs; IgD$^+$ CD27$^+$), resting switched MBCs (swMBC; IgD$^-$ CD27$^+$CD21$^+$), activated swMBC (IgD$^+$ CD27$^+$ CD21$^+$), and double negative B cells (DN; IgD$^-$ CD27$^-$). B) The percentage of type 1, 2, and 3 DN cells among all DN cells. Results are shown for patients who recovered from non-severe (n = 8) and severe (n = 5) COVID-19. $^*$ P < 0.05.
(TIF)

**S2 Fig. Enhanced view of the gating of spike-specific B cells.** The plots show separation of spike-specific B cells from the main cluster of non-specific B cells in the bottom left corner, as well as from B cells reactive with one of the tetramers but not both.
(TIF)

**S3 Fig. Percentage of spike-specific B cells among the major B cell subsets in patients who recovered from non-severe and severe COVID-19.** The percentage of naïve B cells (NBC; IgD$^+$ CD27$^-$), unswitched memory B cells (MBCs; IgD$^+$ CD27$^+$), resting switched MBCs

(swMBC; IgD⁻ CD27⁺ CD21⁺), activated swMBC (IgD⁺ CD27⁺ CD21⁻), and double negative B cells (DN; IgD⁻ CD27⁻) is shown side-by-side for non-RBD-specific (S+RBD-) B cells (left) and RBD-specific (S+RBD+) B cells (right). Results are shown for patients who recovered from non-severe (n = 7) and severe (n = 5) COVID-19.
(TIF)

**S4 Fig. Composite UMAPs for all intracellular and surface markers included in this study.** The plot in the bottom right shows the overlay of all non-RBD-specific (S+RBD-) and RBD-specific (S+RBD+) B cells onto the UMAP.
(TIF)

**S5 Fig. Expression of activation markers in spike-specific and all IgG⁺ B cells in individuals recovered from non-severe and severe COVID-19.** The percentage of CD80, Ki-67, and CD95 (top row) and T-bet, FcRL5, CD11c, and CD21 (bottom row) is shown for spike-specific (S+) IgG⁺ B cells and all IgG⁺ B cells in individuals who experienced non-severe (n = 7) or severe (n = 5) COVID-19. * P < 0.05; ** P < 0.01.
(TIF)

**S6 Fig. Representative flow cytometry plots of the percentage of T-bet⁺ and FcRL5⁺ spike-specific IgG⁺ B cells in individuals recovered from non-severe and severe COVID-19.**
(TIF)

**S7 Fig. Correlation between technical flow cytometry replicates.** Shown are the percentages of spike-specific B cells that express T-bet, FcRL5, CD11c, and CD21 in two technical replicates, one from a non-severe case (pink) and one from a severe case (blue), that were processed and analyzed independently and blinded on separate days. Two data points (pink, ~ 25%) were overlapping and were changed slightly for visualization purposes.
(TIF)

**S8 Fig. Distribution of major B cell subsets in recovered COVID-19 patients at 2–3, 4–5, and 20–23 weeks post-symptom onset.** A) The median distribution of B cell subsets in recovered COVID-19 patients. B) The percentage of naïve B cells (NBC; IgD⁺ CD27⁻), unswitched memory B cells (MBCs; IgD⁺ CD27⁺), resting switched MBC (swMBCs; IgD⁺ CD27⁺ CD21⁺), activated swMBCs (IgD⁻ CD27⁺ CD21⁻), and double negative B cells (DN; IgD⁻ CD27⁻). In all graphs, results are shown for samples collected 2–3 (n = 9), 4–5 (n = 7), and 20–23 (n = 7) weeks post-symptom onset.
(TIF)

**S9 Fig. Percentage of spike-specific B cells among the major B cell subsets in recovered COVID-19 patients 2–3, 4–5, and 20–23 weeks post-symptom onset.** The percentage of naïve B cells (NBC; IgD+CD27-), unswitched memory B cells (MBCs; IgD⁺ CD27⁺), activated switched MBCs (swMBC; IgD⁻ CD27⁺ CD21⁻), resting swMBC (IgD⁺ CD27⁺ CD21⁺), and double negative B cells (DN; IgD⁻ CD27⁻) is shown side-by-side for non-RBD-specific (S+RBD-) B cells (left) and RBD-specific (S+RBD+) B cells (right). In all graphs, results are shown for samples collected 2–3 (n = 9), 4–5 (n = 7), and 20–23 (n = 7) weeks post-symptom onset. * P < 0.05; ** P < 0.01; *** P < 0.001.
(TIF)

**S10 Fig. Expression of activation markers in all IgG⁺ B cells shortly after recovery and five months post-symptom onset.** In all graphs, results are shown for samples collected 2–3 (n = 9), 4–5 (n = 7), and 20–23 (n = 7) weeks post-symptom onset. * P < 0.05.
(TIF)

**S11 Fig. Quality control of purified S and RBD proteins by SDS-PAGE. A)** 800 ng of spike, RBD, and BSA was run on a 4–12% Bis-Tris gel and stained using Imperial Protein Stain. **B)** The full uncropped image of the gel shown in panel A.
(TIF)

**S1 Table. COVID-19 patient characteristics.**
(PDF)

**S2 Table. Reagents and antibodies used for spectral flow cytometry.**
(PDF)

## Acknowledgments

The following reagent was produced under HHSN272201400008C and obtained through BEI Resources, NIAID, NIH: Vector pCAGGS Containing the SARS-Related Coronavirus 2, Wuhan-Hu-1 spike glycoprotein receptor binding domain (RBD), NR-52309. SARS-CoV-2 spike HexaPro was a gift from Dr. Jason McLellan (Addgene plasmid #154754). The plasmid encoding BirA was a kind gift from Dr. Gavin Wright (Addgene plasmid # 32408).

## Author Contributions

**Conceptualization:** Raphael A. Reyes, Evelien M. Bunnik.

**Formal analysis:** Raphael A. Reyes, Sebastiaan Bol.

**Funding acquisition:** Evelien M. Bunnik.

**Investigation:** Raphael A. Reyes, Kathleen Clarke, S. Jake Gonzales, Angelene M. Cantwell, Sebastiaan Bol.

**Resources:** Gabriel Catano, Robin E. Tragus, Thomas F. Patterson.

**Supervision:** Evelien M. Bunnik.

**Visualization:** Evelien M. Bunnik.

**Writing – original draft:** Raphael A. Reyes, Sebastiaan Bol, Evelien M. Bunnik.

**Writing – review & editing:** Raphael A. Reyes, Kathleen Clarke, S. Jake Gonzales, Angelene M. Cantwell, Rolando Garza, Sebastiaan Bol, Evelien M. Bunnik.

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
