## [Decision Letter · Decision Letter 0]

1 Dec 2021

PONE-D-21-36521SARS-CoV-2 spike-specific memory B cells express markers of durable immunity after non-severe COVID-19 but not after severe diseasePLOS ONE

Dear Dr. Bunnik,

Thank you for submitting your manuscript to PLOS ONE. After careful consideration, we feel that it has merit but does not fully meet PLOS ONE’s publication criteria as it currently stands. Therefore, we invite you to submit a revised version of the manuscript that addresses the points raised during the review process.

We look forward to receiving your revised manuscript.

Kind regards,

Menno C van Zelm, PhD

Academic Editor

PLOS ONE

Journal Requirements:

" ext-link-type="uri" xlink:type="simple">https://journals.plos.org/plosone/s/file?id=ba62/PLOSOne_formatting_sample_title_authors_affiliations.pdf"

This work was supported by a COVID-19 pilot award from the UT Health Long School of Medicine (10009547 to E.M.B.). R.A.R. was supported by Translational Science Training award TL1 TR002647. Data were generated in the Flow Cytometry Shared Resource Facility, which is supported by UT Health, NIH-NCI P30 CA054174-20 (CTRC at UT Health) and UL1 TR001120 (CTSA grant). The following reagent was produced under HHSN272201400008C and obtained through BEI Resources, NIAID, NIH: Vector pCAGGS Containing the SARS-Related Coronavirus 2, Wuhan-Hu-1 spike glycoprotein receptor binding domain (RBD), NR-52309. SARS-CoV-2 spike HexaPro was a gift from Dr. Jason McLellan (Addgene plasmid #154754). 

This work was supported by a COVID-19 pilot award from the UT Health Long School of Medicine (10009547 to E.M.B.). R.A.R. was supported by Translational Science Training award TL1 TR002647. Data were generated in the Flow Cytometry Shared Resource Facility, which is supported by UT Health, NIH-NCI P30 CA054174-20 (CTRC at UT Health) and UL1 TR001120 (CTSA grant). The funders had no role in study design, data collection and analysis, decision to publish, or preparation of the manuscript.

Additional Editor Comments:

The manuscript provides a detailed description of the B-cell responses SARS-CoV-2 after mild or severe COVID-19. As indicated by reviewer 1, the patient groups are not very large, but the work is very detailed and confirms and extends previous observations. I recommend a minor revision in which all of reviewer 1' s comments are addressed. In addition, I have 2 minor comments that need to be addressed:

- Please state how many spike and RBD specific cells were gated in each sample. Perhaps a median/range? This is important for the reader to assess whether there has been a true population (30 events), and how reliable subsetting of this population is in each sample.

- Please show pairwise comparisons of T-bet, FcRL5 and CD11c of total B cells in Fig 4 alongside this analysis on Ag-specific cells. Are these markers only different on antigen-specific cells? Or is there a general shift on all B cells? the first is purely related to the response. The latter (all B cells) could indicate effects of severe infection on the total immune system. It is very well possible that the 'cytokine storm' affects the B-cell compartment independent of antigen-specificity

Please make sure to discuss the outcomes of these 2 changes in the Discussion as well.

Reviewers' comments:

Reviewer's Responses to Questions

**Comments to the Author**

1. Is the manuscript technically sound, and do the data support the conclusions?

Reviewer #1: Partly

2. Has the statistical analysis been performed appropriately and rigorously? 

Reviewer #1: Yes

3. Have the authors made all data underlying the findings in their manuscript fully available?

Reviewer #1: Yes

4. Is the manuscript presented in an intelligible fashion and written in standard English?

Reviewer #1: Yes

5. Review Comments to the Author

Reviewer #1: The revised manuscript includes some minor edits to the text to clarify that the differences seen between non-severe and severe COVID-10 are subtle. However, the authors have not changed the title of the manuscript which states that spike-specific memory B cells do not express markers of durable immunity after severe disease.

Considering that the only significant differences between spike-specific memory B cells between the groups is a small decrease in T-bet and Fcrl5 expression in severe cases this title overstates the findings of the manuscript. Any conclusion about the duration of the spike-specific memory B cell response would require a longitudinal study of the severely infected patient cohort which the authors were unable to perform due to a lack of success in recruited recovered severely infected patients for a follow up blood draw. Therefore, it is not appropriate to imply in the title that the memory B cell response is not durable after severe infection. For this manuscript to be appropriate for publication the title should be revised to more accurately represent their findings that there is decreased expression of T-bet and Fcrl5 following severe infection rather than making a broader statement about the duration of immune memory.

6. PLOS authors have the option to publish the peer review history of their article (what does this mean?). If published, this will include your full peer review and any attached files.

Reviewer #1: No

---

## [Author Response · Author response to Decision Letter 0]

3 Dec 2021

We would like to thank the editor and the reviewer for their feedback. Please find below a point-by-point reply to comments. Changes in the manuscript are highlighted in red text.

Editor comments:

The manuscript provides a detailed description of the B-cell responses SARS-CoV-2 after mild or severe COVID-19. As indicated by reviewer 1, the patient groups are not very large, but the work is very detailed and confirms and extends previous observations. I recommend a minor revision in which all of reviewer 1' s comments are addressed. In addition, I have 2 minor comments that need to be addressed:

- Please state how many spike and RBD specific cells were gated in each sample. Perhaps a median/range? This is important for the reader to assess whether there has been a true population (30 events), and how reliable subsetting of this population is in each sample.

In the results section, we already mentioned that donors had between 42 and 450 spike-specific B cells and that a median of 22% was RBD-specific. We have added the median number of spike-specific B cells (median n = 121), as well as the median number and range for RBD-specific B cells (median n = 18, range 7 – 84).

- Please show pairwise comparisons of T-bet, FcRL5 and CD11c of total B cells in Fig 4 alongside this analysis on Ag-specific cells. Are these markers only different on antigen-specific cells? Or is there a general shift on all B cells? the first is purely related to the response. The latter (all B cells) could indicate effects of severe infection on the total immune system. It is very well possible that the 'cytokine storm' affects the B-cell compartment independent of antigen-specificity

We have added an additional panel (E) to Figure 4 where we show an analysis of T-bet, FcRL5, and CD11c expression in total IgG+ B cells. We detected a statistically significant difference in expression between the two groups for FcRL5, although this difference was smaller (1.4-fold) than what we observed for spike-specific B cells (2.0-fold). We also added a statement that these results suggest that the changes we observed in spike-specific B cells are most likely directly related to the immune response to SARS-CoV-2.

Please make sure to discuss the outcomes of these 2 changes in the Discussion as well.

The discussion now includes a statement on the reliability of our conclusions based on the detection of meaningful numbers of spike-specific B cells. In addition, we added a statement discussing the similarity of T-bet and FcRL5 expression in all B cells between non-severe and severe groups, suggesting that there is not a global change to the B cell compartment as a result of the infection. 

Reviewer #1 comments: 

The revised manuscript includes some minor edits to the text to clarify that the differences seen between non-severe and severe COVID-10 are subtle. However, the authors have not changed the title of the manuscript which states that spike-specific memory B cells do not express markers of durable immunity after severe disease.

Considering that the only significant differences between spike-specific memory B cells between the groups is a small decrease in T-bet and Fcrl5 expression in severe cases this title overstates the findings of the manuscript. Any conclusion about the duration of the spike-specific memory B cell response would require a longitudinal study of the severely infected patient cohort which the authors were unable to perform due to a lack of success in recruited recovered severely infected patients for a follow up blood draw. Therefore, it is not appropriate to imply in the title that the memory B cell response is not durable after severe infection. For this manuscript to be appropriate for publication the title should be revised to more accurately represent their findings that there is decreased expression of T-bet and Fcrl5 following severe infection rather than making a broader statement about the duration of immune memory.

We accept the reviewers concern about the title and have changed it to reflect our findings more accurately:

“SARS-CoV-2 spike-specific memory B cells express higher levels of T-bet and FcRL5 after non-severe COVID-19 as compared to severe disease“

---

## [Editor Report · Decision Letter 1]

9 Dec 2021

SARS-CoV-2 spike-specific memory B cells express higher levels of T-bet and FcRL5 after non-severe COVID-19 as compared to severe disease

PONE-D-21-36521R1

Dear Dr. Bunnik,

We’re pleased to inform you that your manuscript has been judged scientifically suitable for publication and will be formally accepted for publication once it meets all outstanding technical requirements.

Kind regards,

Menno C van Zelm, PhD

Academic Editor

PLOS ONE
---

## [Editor Report · Acceptance letter]

13 Dec 2021

PONE-D-21-36521R1 

SARS-CoV-2 spike-specific memory B cells express higher levels of T-bet and FcRL5 after non-severe COVID-19 as compared to severe disease 

Dear Dr. Bunnik:

I'm pleased to inform you that your manuscript has been deemed suitable for publication in PLOS ONE. Congratulations! Your manuscript is now with our production department. 

Kind regards, 

on behalf of

Dr. Menno C van Zelm 

Academic Editor

PLOS ONE